# Enzymatic Glycerolysis of Palm Kernel Olein-Stearin Blend for Monolaurin Synthesis as an Emulsifier and Antibacterial

**DOI:** 10.3390/foods11162412

**Published:** 2022-08-11

**Authors:** Ngatirah Ngatirah, Chusnul Hidayat, Endang S. Rahayu, Tyas Utami

**Affiliations:** 1Study Program of Food Science, Faculty of Agricultural Technology, Universitas Gadjah Mada, Yogyakarta 55281, Indonesia; 2Department of Agricultural Product Technology, Institut Pertanian Stiper, Yogyakarta 55283, Indonesia; 3Department of Food and Agricultural Product Technology, Faculty of Agricultural Technology, Universitas Gadjah Mada, Yogyakarta 55281, Indonesia

**Keywords:** palm kernel olein-stearin blend, enzymatic glycerolysis, monolaurin, emulsifier-antibacterial characteristics, antibacterial lipid

## Abstract

Monolaurin is a monoacylglycerol, which can act as an emulsifier and antibacterial. Palm kernel oil is a monolaurin raw material that can be fractionated into palm kernel olein (PKOo) and palm kernel stearin (PKS). Therefore, this study prepares monolaurin through enzymatic glycerolysis of the PKOo-PKS blend. The effects of enzyme concentration, molar ratio of oil to glycerol, solvent to oil ratio, and reaction temperature on the products of glycerolysis were investigated. The best conditions were selected for further production, purification, and characterization of the monolaurin. The results showed that the best glycerolysis condition was obtained with an enzyme concentration of 10% *w*/*w*, an oil–glycerol molar ratio of 1:4, a solvent–oil ratio of 2:1 *v*/*w*, and a glycerolysis temperature of 40 °C with a stirring speed of 600 rpm based on the monoacylglycerol (MAG) concentration. The identification of the sample with FTIR and NMR indicated that the purified glycerolysis product is the monolaurin. The thermal analysis showed a large endothermic peak with a melting point of 35.56 °C. The purified monolaurin has a HLB value of 5.92, and an emulsion capacity and stability of 93.66 ± 1.85% and 89.54 ± 3.36%, respectively. The minimum inhibitory concentration (MIC) of the monolaurin for *Escherichia coli* FNCC 0091 and *Staphylococcus aureus* FNCC 0047 were at 500 ppm, and 100 ppm for *Bacillus subtilis* FNCC 0060.

## 1. Introduction

Monolaurin is naturally found in breast milk, at 5.8% of fat. With two hydroxyls and one lauryl group, monolaurin serves as an essential fat and immune system booster for infants [1]. Based on its classification, this monoglyceride is considered a non-ionic emulsifier due to the presence of hydrophilic and hydrophobic groups in its molecular structure. [2]. It is used as an emulsifier in the food and pharmaceutical industry. Furthermore, this compound has the ability to act as an antimicrobial agent for food preservatives [3]. Monolaurin is a nontraditional antimicrobial agent with better antimicrobial activities but no health risks to consumers; however, its use as a preservative in the food industry is still limited [4]. Monolaurin is also a valuable product due to its many biologically valuable properties and the benefits it provides to human health.

The monolaurin sources are lauric acid (C_12_H_24_O2), methyl lauric (C_13_H_26_O_2_), coconut, and palm kernel oils (PKO). Lauric acid and methyl lauric can only be obtained from hydrolysis or transesterification from coconut oil or PKO. Therefore, the sources of monolaurin with the most potential are coconut and PKO, the latter of which being a by-product of crude palm oil (CPO) processing with higher productivity advantages than other sources in Indonesia. Monolaurin production from PKO is achieved through glycerolysis reaction. An advantage of the glycerolysis process that warrants its study is that it is simple to carry out, as fat or oil can be used directly as a substrate without the need to separate free fatty acids [5].

As a monolaurin raw material, PKO has several limitations, such as the variations in olein and stearin content, due to the processing condition and storage condition of the kernel. PKO can be fractionated further to produce 60–70% palm kernel olein (PKOo) and 30–40% palm kernel stearin (PKS), both of which contain 39.7–48% and 56–59.7% lauric acid, respectively [6]. PKOo and PKS potentially serve as the raw material for monolaurin synthesis based on their lauric acid composition. Despite the accessibility of PKS being lower than PKOo, its lauric acid content is lower. Thus, blending PKOo and PKS is required to increase the amount of lauric acid in the fat blend. The PKOo-PKS blend also changes the trilaurin and lauric acid content significantly, with the lauric acid content at the sn-2 position. A previous study found that the PKOo-PKS blend in ratios of 40:60 produced trilaurin and lauric acid values of 24.58 ± 0.56% and 57.01 ± 0.15%, respectively. The ratio of PKOo-PKS at 40:60 have 50.71% lauric acid at sn-2 position [7].

The commercial productions of monolaurin are carried out through lauric acid esterification with chemical catalysts. There are several limitations to these chemical catalysts such as high temperatures (220–260 °C), high consumption of energy, not being environmentally friendly, and darker products [8]. Alternatively, some studies conducted the enzymatic glycerolysis process, where fat or oil can be used as a direct substrate without separating free fatty acids using a mild temperature condition [5]. Enzymatic glycerolysis can be used to produce specific monoglyceride in terms of fatty acid position due to the position specificity of lipase [5]. Therefore, monolaurin synthesis can be developed using enzymatic glycerolysis.

Limited studies have investigated the enzymatic glycerolysis of the PKOo-PKS blend for monolaurin synthesis using Lipozyme RM IM. The synthesis of monoacylglycerol (monolaurin) was accomplished through the glycerolysis of coconut oil and crude glycerol in ethanol at an 8:1 molar ratio of glycerol to oil at 45 °C for 36 h, and an enzyme of 20 wt% with an initial water activity of 0.53 [9]. Zha et al. found that the optimum conditions for coconut oil glycerolysis at 50 °C in the microemulsion system were the coconut oil to glycerol molar ratio 1:4, the concentration of Novozym 435 and sodium (bis-2-ethyl-hexyl) sulfosuccinate 8 and 16%, respectively [10]. The separation of monoacylglycerol or monolaurin can be achieved using column chromatography with silica gel as the stationary phase and a mixture of solvents *n*-hexane and ethyl acetate, which are gradually eluted [11,12], or the solvent method using *n*-hexane and hydroalcoholic solution [13]. However, there is limited information on the various factors that can impact the effectiveness of PKOo-PKS blend glycerolysis to produce monolaurin, including the concentration of enzyme, the molar ratio of oil to glycerol, the solvent to oil ratio, and the temperature reaction.

This study aimed to produce and characterize monolaurin from the PKOo-PKS blend through enzymatic glycerolysis. The emulsifying and antibacterial characteristics of purified monolaurin were studied.

## 2. Materials and Methods

### 2.1. Materials

The PKOo, PKS, and refined glycerin were obtained from PT Wilmar, Indonesia. Lypozyme RM IM and molecular sieves were purchased from Sigma-Aldrich (St. Louis, MO, USA). Hexane, ethanol, tert-butanol, nutrient broth and nutrient agar were purchased from Merck KGaA (Darmstadt, Germany), while filter paper discs (6 mm diameter) were purchased from Oxoid. *Escherichia coli* FNCC 0091, *Staphylococcus aureus* FNCC 0047, and *Bacillus subtilis* FNCC 0060 were obtained from Food and Nutrition Culture Collection, Center for Food and Nutrition Studies Universitas Gadjah Mada, Indonesia.

### 2.2. Preparation of Fat Blends

The refined PKOo and PKS were heated separately at 70 °C for 30 min. Then, the liquefied PKOo and PKS were mixed at a ratio of 40:60 (*w*/*w*), homogenized at 70 °C for 15 min using a hotplate (Thermo Scienctific Cimarec, Waltham, MA, USA) with magnetic stirrer, and stored in a refrigerator for further experiments.

### 2.3. Enzymatic Glycerolysis of PKOo-PKS Blend

Enzymatic glycerolysis was carried out in a flask placed in a water bath. The reaction mixture was composed of a PKOo-PKS blend (40:60 *w*/*w*) 10 g, tert-butyl alcohol, glycerol, molecular sieve 12% of glycerol, and lipase. The mixture was stirred by a stirrer (RW 20, IKA, Staufen, Germany), at 200 to 600 rpm. The experimental details consist of enzyme concentrations of 1% (30 U/g), 3% (90 U/g), 5% (150 U/g), 10% (300 U/g), and 15% (450 U/g) *w*/*w* of the fat. The molar ratio of oil to glycerol was from 1:2 to 1:10; the ratio of tert-butanol to oil was from 1.5:1 to 3:1 *v*/*w*, and the temperature range from 40 to 60 °C. After 24 h reaction, the product was separated from enzyme and glycerol using centrifugation (Hettich EBA 200, Tuttlingen, Germany) at 3000 rpm for 5 min and then separated from the solvent using a rotary evaporator (Heidolph, Schwabach, Germany). The composition of monoacylglycerol (MAG), diacylglycerol (DAG), and triacylglycerol (TAG) was analyzed using gas chromatography (shimadzu GC-14B). The best conditions process of glycerolysis was used to produce monolaurin.

### 2.4. Production and Isolation of Monolaurin

Monolaurin production was carried out under the best glycerolysis conditions according to the procedure described in previous studies [14]. A 100 g PKOo-PKS blend in ratio of 40:60 *w*/*w* was heated at 70 °C for 30 min. The product was melted and mixed in a 1:4 ratio with glycerol. Tert-butanol was then added to the mixture in a 2:1 oil-to-tert-butanol ratio. The molecular sieve added 12% wt glycerol. Furthermore, the mixture was mixed at a speed of 600 rpm in the batch stirred tank reactor at 40 °C. The mixture was then incubated for 24 h with Lipozyme RM IM containing 10% (300 U/g) *w*/*w* oil. The product separation from the enzyme and glycerol was carried out through centrifugation (Hettich EBA 200, Germany) at 3000 rpm for 5 min and then extracted from the solvent using a rotary evaporator (Heidolph, Germany). Monolaurin isolation was conducted by Nitbani et al. [13] with modification. The glycerolysis products were dissolved in an hydroalcoholic (ethanol:water = 8:2), with a ratio of 1:9 *v*/*v*, and were then put in the refrigerator for 24 h. MAG was dissolved in the hydroalcoholic phase, while DAG and TAG were solidified and filtered using vacuum filtration. The filtrate was added *n*-hexane with a ratio of 1:3 *v*/*v* and stood in the separating funnel for 24 h to form 2 layers. The bottom layer was taken, and the hydroalcoholic was removed using a rotary evaporator (Heidolph, Germany) to obtain monolaurin. The purified monolaurin was weighed and identified using FTIR (ABB MB3000, Clairet Scientific Ltd., Northampton, UK), NMR (JNM-ECZ500R, 500 MHz super conductive magnets), and DSC (DSC-60 Plus Shimadzu, Kyoto, Japan). Its capability as an emulsifier and its antibacterial activity were also analyzed.

### 2.5. Determination of Emulsion Capacity and Stability

Emulsion capacity and stability were determined according to Cano-Medina et al. [15] with modification. Five milliliters of monolaurin solution (1 percent *w*/*v*) were homogenized with five milliliters of soybean oil. The emulsion was centrifuged for 5 min at 1100 rpm. The height of the emulsified layer and the total contents of the tube were both measured. The emulsion capacity (EC) was calculated as:$$\mathrm{EC}=\frac{\mathrm{Height}\;\mathrm{of}\;\mathrm{emulsified}\;\mathrm{layer}\;\mathrm{in}\;\mathrm{the}\;\mathrm{tube}}{\mathrm{Height}\;\mathrm{of}\;\mathrm{the}\;\mathrm{total}\;\mathrm{contents}\;\mathrm{in}\;\mathrm{the}\;\mathrm{tube}}\times 100\%$$

Emulsion stability (ES) was determined by heating the emulsion at 80 °C for 30 min before centrifuging at 1100 rpm for 5 min.
$$\mathrm{ES}=\frac{\mathrm{Height}\;\mathrm{of}\;\mathrm{emulsified}\;\mathrm{layer}\;\mathrm{after}\;\mathrm{heating}}{\mathrm{Height}\;\mathrm{of}\;\mathrm{the}\;\mathrm{emulsified}\;\mathrm{layer}\;\mathrm{before}\;\mathrm{heating}}\times 100\%$$

### 2.6. Antibacterial Assay with the Paper Disc Diffusion Method

The antibacterial assay was determined according to Priscilia et al. [16], with modification. First, the bacterial suspension of *Escherichia coli* strain, *Staphylococcus aureus* strain, and *Bacillus subtilis* strain was prepared. Then, 20 g of nutrient agar was dissolved in 1 L of distilled water and then sterilized at a temperature of 121 °C for 15 min. The nutrient agar solution was poured into the sterile Petri dish. After it was solidified, the nutrient agar media plate was inoculated with 0.1 mL bacterial suspension. The blank paper discs were then dipped in a monolaurin solution with concentrations of 100, 500, 1000, 2500, and 5000 ppm prepared with aquadest solvents and ethanol. They were placed on the plate agar and incubated for 24 h at 37 °C. The monolaurin solution diffused into the agar and inhibited the growth of the tested microorganism. Then, the diameters of clear zones were measured with millimeter (mm) units.

### 2.7. Statistical Analysis

A single factor completely randomized design was used as the experimental design (CRD). The experiment and analysis of samples were conducted with duplicates with the exception of the antibacterial assay which was conducted with triplicates. The data were analyzed by one-way analysis of variance and a significant difference between the treatment followed by a Duncan Multiple Range Test (DMRT) level 5%. The results were expressed as means ± standard deviation.

## 3. Results and Discussion

### 3.1. Effect of Reaction Conditions on Glycerolysis of PKOo-PKS Blend

Figure 1a–d shows the effect of enzyme concentration, the molar ratio of oil to glycerol, the ratio of solvent to glycerol, and the temperature on glycerolysis of the PKOo-PKS blend.

From Figure 1a, it can be seen that the composition of MAG, DAG, and TAG at different enzyme concentrations (1–15% *w*/*w* of oil) did not show a significant difference. TAG conversion mainly consists of DAG (63.89–68.71%) and slightly of MAG (0.72–0.59%). This was due to the specific sn-1,3 enzyme, which decomposed fatty acids in TAG at the sn 1 and 3 positions to produce 1,3-DAG and 1,2-DAG, respectively. Furthermore, the DAG values were high because it was assumed that the lipozyme RM IM enzyme did not hydrolyze DAG into MAG. The molar ratio of oil to glycerol, amount of solvent, reaction temperature, and stirring speed all had an effect on glycerolysis. The reaction system can form DAG when glycerol availability is limited. The TAG glycerolysis is a two-way reaction, where excess glycerol directs the reaction equilibrium towards the formation of MAG.

The MAG increased with higher lipase concentration due to the more significant number of active sites. However, the total MAG content was not significantly increased when the enzyme was elevated from 10% *w*/*w* of oil. This is due to catalytic activity saturation when the addition of lipase was greater than 10% *w*/*w* of oil. Diao et al. reported that when the enzyme concentration was greater than 14% (*w*/*w*) of the enzyme to lard substrate ratio, the contents of total DAG and 1,3 DAG were not significantly increased [17]. Some studies observed that a poor blend of the reaction mixtures can be caused by a high amount of enzyme in the glycerolysis reaction, leading to limited mass transfer [17]. Solaesa et al. reported that with a glycerol:oil mole ratio of 3:1, the MAG yield increased in the glycerolysis of sardine oil using Lipozyme 435 at loads of 5 and 10% wt percent based on reactant weight [18]. Given the efficiency of biocatalysts, an enzyme concentration of 10% (*w*/*w*) was chosen for future experiments.

The glycerolysis process in Figure 1b was conducted at 50 °C, the ratio of tert-butanol and oil was 1.5:1 with stirring 200 rpm, an enzyme concentration of 10% *w*/*w* of oil, and the molar ratio of oil to glycerol was 1:2 to 1:10. As shown in Figure 1b, the total MAG content was not significantly different with increasing glycerol concentration. After shifting to an oil to glycerol molar ratio of 1:4, the MAG content increased initially, followed by a decreasing pattern. A ratio of 1:4 produces the highest MAG (0.59%), although it is not significantly different. The molar ratio of oil to glycerol over 1:4 produced a lower MAG. This is due to the fact that a higher glycerol concentration can result in higher viscosity of the reaction mixture and significant mass transfer resistance [17]. In the longer-term, excess glycerol inhibits the enzyme activity due to its polar nature as it can strip the water out from the immobilized enzyme which eventually could lead to its inactivation [19]. The higher molar fraction of TAG compared to glycerol enhanced the synthesis of dilaurin. Furthermore, due to the higher production of DAG, free fatty acids did not react to the glycerol. During glycerolysis reactions, the majority of the glycerol reacted with fatty acids derived from TAG to produce monoacylglycerol [20]. Theoretically, the stoichiometry of the glycerolysis reaction requires a glycerol to fat (oil) molar ratio of 1:2 to provide 3 moles of MAG during this process [17,21]. However, the yield of monoacylglycerol depends on the equilibrium conditions, due to several factors [22]. The substrate molar ratio of oil to glycerol was essential in determining the chemical balance and reaction speed of the glycerolysis. Furthermore, the amount of glycerol plays an important role in determining the progress of TAG conversion and the composition of the reaction products [23]. In this reaction, MAG and DAG were high based on the excess and limited use of glycerol, respectively [24]. According to the equilibrium law, an increase in the glycerol content shifted the balance towards the production of MAG [21] and a molar ratio of oil to glycerol of 1:4 was selected for further experiments.

The effects of solvent to oil ratio on the enzymatic glycerolysis reaction of palm kernel olein-stearin blend are presented in Figure 1c. Subsequently, the total MAG content increased with the increasing reaction solvent to oil ratio until 2:1 (*v*/*w*), and then decreased at ratio 3:1 (*v*/*w*). Increasing the solvent to oil ratio (3:1 *v*/*w*) increases and decreases the content of DAG and MAG, respectively. The conversion rates increased with the solvent to substrate ratio from 1.5:1 to 2:1 *v*/*w* [25]. The presence of a solvent (tert-butanol) helped both reactants to diffuse to the enzyme’s active sites [26]. The addition of more solvent increased the solubility of the substrates, mixture homogeneity and stability, and substrate diffusivity due to the change in the fat and glycerol mixture. On the other hand, higher solvent amount may reduce the concentration of substrates, which in turn influences the reaction rate according to the Michaelis–Menten equation [25]. Therefore, a solvent to oil ratio of 2:1 *v*/*w* was selected for further experiments.

The effects of temperature on the enzymatic glycerolysis reaction of palm kernel olein-stearin blend are presented in Figure 1d. The results showed that temperature had a considerable influence on the content of MAG and DAG. The MAG in the samples at 40°C was significantly higher than at 45, 50, 55, and 60 °C. The MAG contents decreased with increasing temperature reaction, until 60 °C (Figure 1d). A higher temperature causes enzyme denaturation [17]. The highest MAG (31.64%) was obtained at a reaction temperature of 40 °C due to Lipozyme RM IM having a maximum activity at 40 °C. Diao et al. found that the optimum temperature for the glycerolysis of lard was 45°C [17]. Skoronski et al. reported that a higher conversion on an aliphatic ester production using Lipozyme^®^ RM IM was obtained at 40 °C, while a larger amount of ester was produced when the reaction was carried out at 30 °C [27]. As the temperature increased, the enzyme’s activity decreased [27]. Shahrin et al. reported that the optimal temperature for the catalyzing activity of Lipozyme RM IM in monolaurin synthesis using lauric acid and crude glycerol as a substrate was performed at 47 °C [28]. Therefore, a temperature of 40 °C was selected for further experiments considering the content of MAG.

Stirring speed significantly influenced the content of MAG in the glycerolysis reaction of the palm kernel olein-stearin blend. For example, at 200 rpm stirring (Figure 1a–c), the content of MAG is low, while when stirring at 600 rpm (Figure 1d) a higher content of MAG was obtained. This is due to the increased contact area of the enzyme and substrate and improved catalytic efficiency at a certain stirring speed. Therefore, the stirring speed should be at 500 rpm during the reaction to yield the highest TAG conversion and DAG content [17]. Based on this study, an enzyme concentration of 10% *w*/*w* (300 U/g), oil to glycerol molar ratio of 1:4, solvent to oil ratio of 2:1 *v*/*w*, temperature reaction of 40 °C, and stirring speed 600 rpm were selected as the best condition for monolaurin production.

### 3.2. Characterization of Monolaurin Isolated from Glycerolysis Product

#### 3.2.1. Analysis of FTIR (Fourier Transform Infra-Red) Monolaurin

The result at Figure 2 showed that after purification, the monolaurin sample was analyzed using FTIR. It indicated that the OH group occurred at wavenumbers of 3368.64 cm^−1^, 1734.03 cm^−1^, 1174.05–993.03 cm^−1^, 2853.21–2922.01 cm^−1^, and 1461.04–1380.46 cm^−1^, respectively, as well as C=O ester, C-O-C, CH stretch vibration, and methyl and methylene group. The rocking vibration from (CH_2_)*n* was also indicated by the wavenumber of 719.77 cm^−1^. According to the FTIR analysis, the sample and standard monolaurin are identical to previous studies.

Widiyarti et al. found a new group at 3224.98 cm^−1^ and 3290.56 cm^−1^. As a result, at a wavenumber of 1730.15 cm^−1^, the asymmetric stretching vibrations of the hydroxyl and carbonyl groups were similar [12]. Sangadah et al. also obtained the C=O ester and OH clusters at 1748 cm^−1^ and 3650–3200 cm^−1^ wavelengths, respectively [29].

#### 3.2.2. Analysis of NMR Monolaurin

The FTIR spectra and NMR analysis (^1^H and ^13^C NMR) confirmed that the synthesized compound is monolaurin, as shown in Figure 3 and Figure 4.

The NMR spectra showed the data spectrum of monolaurin as follows: ^1^H-NMR (500 MHz, Cloroform-D) δ 4.94 (d, *J* = 8.7 Hz, 1H, OH), 4.16 (s, 2H, -(OH)CH-CH_2_ -OCO-), 4.07 (d, *J* = 5.6 Hz, 1H, -CH_2_-(OH)CH-CH_2_-), 3.86 (qd, *J* = 5.8, 3.4 Hz, 1H, CH_2_- (OH)CH-CH_2_-), 3.61 (td, *J* = 10.8, 3.8 Hz, 1H, CH_2_- (OH)CH-CH_2_-), 3.56–3.45 (m, 1H, -CH_2_ -(OH)CH-CH_2_-), 2.27 (h, *J* = 7.6 Hz, 2H, CH_2_ of C2/O=CCH_2_), 1.60–1.51 (m, 2H, CH_2_ of C3/O=CCH_2_CH_2_), 1.31–1.18 (m, 16H, 8CH_2_ of C4-11), 0.82 (t, *J* = 6.9 Hz, 3H, CH_3_ of C12). 

The ^13^C-NMR spectra reported the data spectrum of monolaurin as follows: ^13^C NMR (126 MHz, CHLOROFORM-D) δ 177.58–174.46 (C=O), 72.47–70.24 (-(OH)CH-CH_2_-OCO-), 65.01 (-(OH)CH-CH_2_-OCO-), 63.61–61.43 (-OH)CH_2_-(OH)CH-CH_2_-), 34.38–34.06 (O=C-CH_2_), 31.96–31.70 (CH_2_CH_2_CH_3_), 29.74–29.67 (2CH_2_), 29.54 (CH_2_), 29.49 (CH_2_), 29.39–29.35 (CH_2_), 29.22–28.97 (2CH_2_), 24.92–24.88 (CH_2_), 22.72–22.63 ((-CH_2_CH_3_), 14.13–14.08 (CH_3_).

The ^1^H-NMR spectrum is used to determine the number and position of protons in synthetic compounds. In contrast, the ^13^C-NMR spectrum is used to determine the amount of carbon (C), methyl carbon, methyl, methylene, metin, or carbonyl ester. Spectrum ^1^H-NMR determines the number of hydrogen atoms and the position of protons in the synthesized compound. Based on spectrum data, the number of hydrogen atoms in the synthesized compounds is 30. In comparison, the ^13^C-NMR spectrum is used to determine the amount of carbon (C), methyl carbon, methylene, metin, or carbonyl ester. For example, in Figure 4, the number of C atoms is 15. Spectrum ^1^H-NMR also shows 30 peaks which indicates the amount of hydrogen atoms in the results of the enzymatic glycerolysis of monolaurin. There is a refractive double bond seen in the range of 1.5–3 ppm in this spectrum. The ^1^H and ^13^C-NMR spectrum obtained in this study was similar to previous studies [12,30,31].

#### 3.2.3. DSC Analysis of Monolaurin

Thermal properties of monolaurin isolated from the product glycerolysis palm kernel olein-stearin blend were tested using Differential Scanning Calorimetry (Figure 5).

Thermal analysis of monolaurin isolated from product glycerolysis palm kernel olein-stearin blend showed significant endothermic peaks, which end at a melting point of about 35.56 °C (Figure 5). The endothermic peak of monolaurin is lower than pure monolaurin, which is about 65.5–66 °C [32,33]. This indicates that monolaurin contains impurities that result in a lower melting point. Commercial monolaurin will have a sharp endothermic peak and a melting point of about 50–56 °C. The melting point is higher than the results of Galuh et al., who obtain a monolaurin melting point of about 30 °C [12].

### 3.3. Characterization of Monolaurin as an Emulsifier and Antibacterial

#### 3.3.1. Physicochemical and Emulsifying Properties

The characteristic and emulsifying properties of monolaurin are shown in Table 1.

Table 1 shows that the synthesis monolaurin has a saponification value of about 193.6 ± 0.83 mg KOH/g. This is close to the commercial monolaurin, which has a saponification value of 200–210 mg KOH/g. The iodine value is close to the commercial monolaurin, with a maximum of 1 (wijs). The melting points are lower compared to commercial monolaurin from 50–56 °C. However, it is higher than the results of Galuh et al., who obtain a monolaurin melting point of about 30 °C [12]. The monolaurin from the synthesis has a high emulsion capacity of 93.66% and stability of 89.54%. Therefore, monolaurin has been widely used in emulsions, specifically microemulsions [34]. Zhang et al. found the emulsion stability of monolaurin was 95% at 2 h and 85% at 24 h [35]. In addition, the HLB monolaurin (HLB 5.97) was similar to commercial monolaurin (HLB 5). It was lower than Park et al.’s value which found the monolaurin HLB to be 7.03 [36]. Hydrophilic and lipophilic characteristics of fatty acid derivatives affect their antibacterial activities according to their incorporation into the bacterial cell membrane [36]. This study’s emulsion type of monolaurin is suitable for w/o emulsion because the HLB value was 5.97. The type and size of the molecules influence the emulsifier from hydrophobic and hydrophilic ends to produce group values or HLB. Emulsifiers with low HLB values (3–6) produce a water-in-oil emulsion, while high HLB (8–18) produces an oil-in-water emulsion [29].

#### 3.3.2. Antibacterial Activity of Monolaurin

Monolaurin can inhibit both gram-negative and positive bacteria. Antibacterial assay on *E. coli*, *B. subtilis*, and *S. aureus* were shown in Table 2. Monolaurin isolated from glycerolysis palm kernel olein-stearin can inhibit the microbes of *E. coli* FNCC 0091, *B. subtilis* FNCC 0060, and *S. aureus* FNCC 0047. Furthermore, *E. coli* represents gram-negative bacteria while *B. subtilis* and *S. aureus* represent gram-positive bacteria. The concentration of monolaurin is directly proportional to the inhibitory activity. Monolaurin dissolved with ethanol solvents shows a tethering effect ranging from 100 ppm for all three types of bacteria to 5000 ppm. The antibacterial activity with aquadest solvents has the best inhibitory result at a concentration of 100 ppm for *B. subtilis* FNCC 0060 and 500 ppm for *E. coli* FNCC 0091 and *S. aureus* FNCC 0047. This means the monolaurin inhibitory activity for gram-positive bacteria is more effective than gram-negative. In addition, *B. subtilis* is more sensitive to monolaurin than *S. aureus* and *E. coli.* This result follows the Zhang et al. study that found the minimum inhibitory concentrations for monolaurin to be 25 μg/mL against *E. coli*, 12.5 μg/mL against *S. aureus*, and 30 μg/mL against *B. subtilis* [4]. They exhibited excellent antibacterial activity against *S. aureus*, while *E. coli* was minimally affected due to the hydrophilic structure of the outer membrane [33].

Jumina et al. found that 1-monolaurin at a 100 mg/mL concentration can inhibit *S. aureus* and *E. coli* with an inhibitory zone diameter of 15.8 mm and 12.7 mm, respectively [3]. The 2-monolaurin compound inhibited the growth of *S. aureus* and *B. cereus* at a minimum concentration of 2500 ppm with an inhibitory zone diameter of 13.75 mm and 10.44 mm, respectively, but there was no inhibition for *E. coli* and *S. typhimurium* [13]. Furthermore, the 1-monolaurin can inhibit *E. coli* and *S. aureus* at a minimum concentration of 500 µg/mL with a 7.5 mm and 10.55 mm inhibitory zone, respectively [30]. Galuh et al. found that *S. aureus* can be inhibited at a concentration of at least 500 ppm with a 7 mm inhibitory zone [12]. The MIC values against *E. coli* and *S. aureus* were 2000 and 250 µg/mL, respectively, while the values in combination with EDTA against *E. coli* and *S. aureus* were 1000 and 125 µg/mL, respectively [37]. For the 24-h test period, monolaurin was bactericidal at a concentration of 100 µg/mL for both *B. subtilis* and *B. cereus* [38]. Amin Zare et al. also reported that monolaurin dissolved in 95% ethanol had a minimal inhibitory concentration for *E coli* >4000 μg/mL while for *S. aureus* it was 128 μg/mL [39]. Sadiq et al. reported that the minimal inhibitory concentration for *S. aureus* ATCC 25923 was 100 μg/mL [40]. Wang et al. reported that the MIC for *S. aureus* and *B. subtilis* grown in nutrient broth was 1.25 mg/mL and 0.63 mg/mL, respectively, while for *E. coli* ≥10 mg/mL [41]. Buňková et al. report that monolaurin has an inhibitory effect on *B. subtilis* CCM 4062 and *S. aureus* CCM 3953 which are grown in nutrient broths with the addition of monolaurin at the lowest concentrations of 25 ppm and 250 ppm, respectively, while in *E. coli* the inhibitory effect of monolaurin occurs at 1500 ppm [42].

The mechanism of inhibition of monolaurin to *E. coli* is as follows: it first crosses the cell membrane and disrupts the normal functioning of DNA, eventually causing cell lysis. The effect of monolaurin activity on DNA is that it inhibits the DNA transcription process and causes a reduction in the synthesis of RNA and proteins, thus causing the cell cycle to stop and eventually causing inhibition of cell division [43]. The process of antimicrobial mechanisms includes the following three aspects: increased membrane permeability and cell lysis, disruption of the electron transport chain and oxidative phosphorylation separating, and inhibition of membrane enzymatic activity and nutrient uptake [44].

Based on this study, monolaurin has been used as an integral consistuent of foodstuffs or as part of packaging material to improve food shelf life and/or inhibit microorganism growth. In terms of trends for the future, consumers prefer naturally occurring antimicrobial compounds or preservatives to synthetic chemicals [41].

## 4. Conclusions

Monolaurin can be synthesized from palm kernel olein-stearin blend and has the properties to be used as a food preservative and emulsifier. Monolaurin is also a valuable product due to its many biologically valuable properties and the benefits it provides to human health. The results showed that the best conditions for monolaurin production were as follows: an enzyme concentration of 10% *w*/*w*, a molar ratio of oil to glycerol 1:4, a ratio of solvent to oil of 2:1 *v*/*w*, a reaction temperature of 40 °C and a stirring speed of 600 rpm. Under these conditions, the content of monoglyceride was 31.64%. The FTIR analysis showed that the sample was identical to the monolaurin standard. Identification with NMR indicated that the compound of purified glycerolysis product was the α-monolaurin. Thermal analysis of the purified monolaurin showed significant endothermic peaks, which ended at a melting point of about 35.56 °C. The purified monolaurin has a MAG content of 83.89 ± 5.52%, HLB value of 5.92, and emulsion capacity and stability of 93.66 ± 1.85% and 89.54 ± 3.36%, respectively. The antibacterial activity with aquadest solvents has a minimum inhibitory concentration of 500 ppm for *E. coli* FNCC 0091 and *S. aureus* FNCC 0047 and 100 ppm for *B. subtilis* FNCC 0060. Therefore, monolaurin has the potential to be developed as a food preservative and supplement for human health.

## Figures and Tables

**Figure 1 foods-11-02412-f001:**
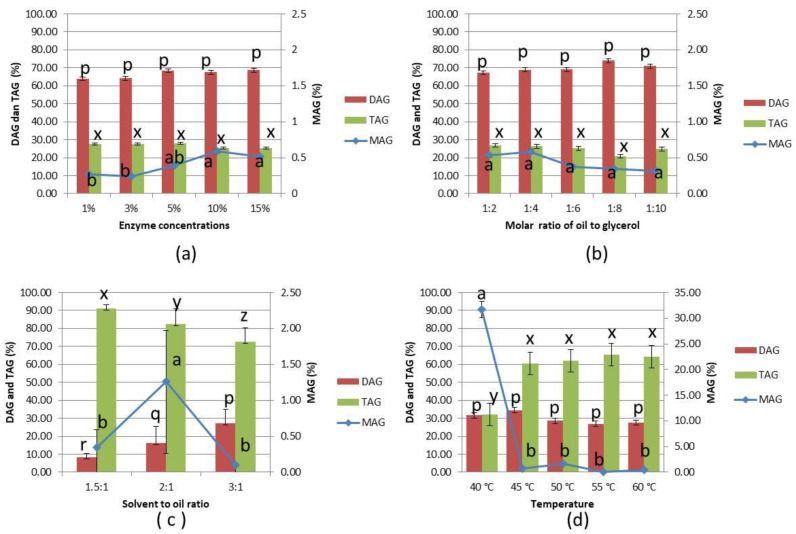
Composition of MAG, DAG, and TAG from the glycerolysis of PKOo-PKS blend at a ratio of 40:60 (*w*/*w*) for 24 h under various conditions: (**a**) enzyme concentration 1–15% *w*/*w*, molar ratio oil to glycerol of 1:4, stirring 200 rpm, tert-butanol to oil ratio of 1.5:1 *v*/*w*, temperature 50 °C, (**b**) molar ratio oil to glycerol of 1:2 to 1:10, enzyme concentration 10%, stirring concentration 200 rpm, tert-butanol to oil ratio of 1.5:1 *v*/*w*, temperature 50 °C, (**c**) solvent to oil ratio of 1.5:1 to 3:1 *v*/*w*, molar ratio oil to glycerol of 1:4, enzyme concentration 10%, stirring 200 rpm, temperature 50 °C, and (**d**) temperature 40–60 °C, molar ratio oil to glycerol of 1:4, enzyme concentration 10%, stirring 600 rpm, solvent to oil ratio of 2:1 *v*/*w*. Error bars refer to the standard deviations obtained from duplicate sample analysis in duplicate experiments. Means in the same indexes with different letters (a–d) differ significantly at *p* < 0.05 by Duncan’s multiple range test.

**Figure 2 foods-11-02412-f002:**
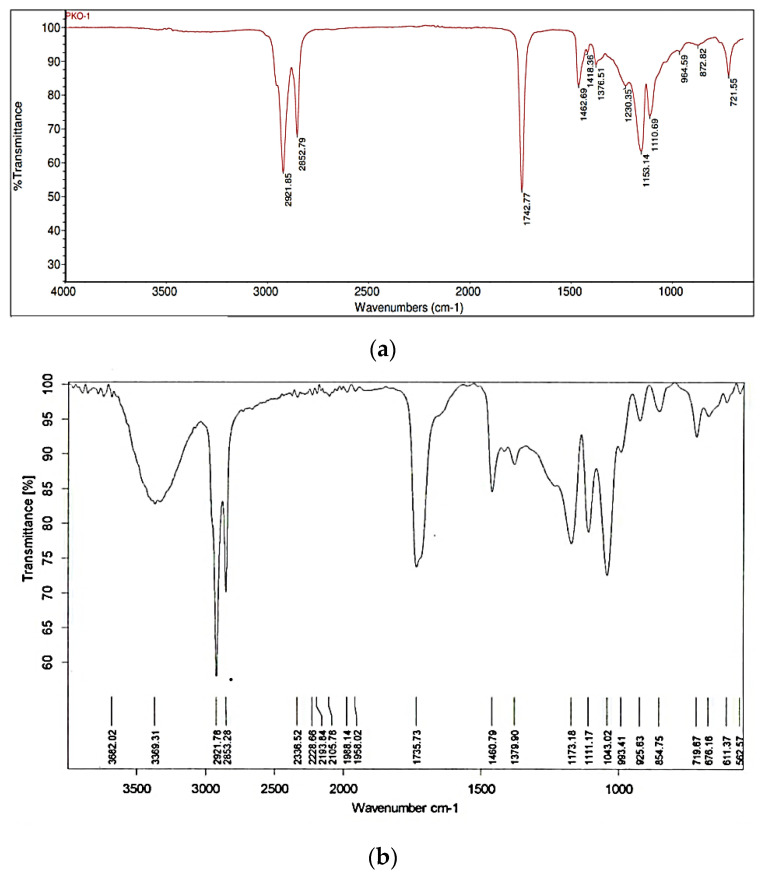
(**a**) FTIR spectra of PKOo-PKS blend at a ratio of 40:60 *w*/*w*, and (**b**) FTIR spectra of purified monolaurin from glycerolysis product.

**Figure 3 foods-11-02412-f003:**
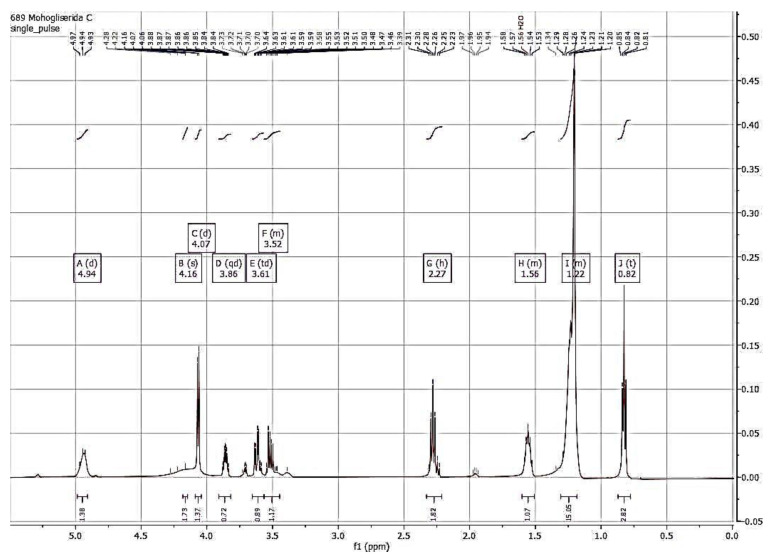
Spectrum ^1^H NMR of monolaurin from glycerolysis reaction of PKOo-PKS blend with a ratio of 40:60 *w*/*w* for 24 h, at 40 °C, molar ratio of oil to glycerol 1:4, solvent to oil ratio of 2:1 *v*/*w*, enzyme concentration of 10% *w*/*w*, and stirring speed 600 rpm.

**Figure 4 foods-11-02412-f004:**
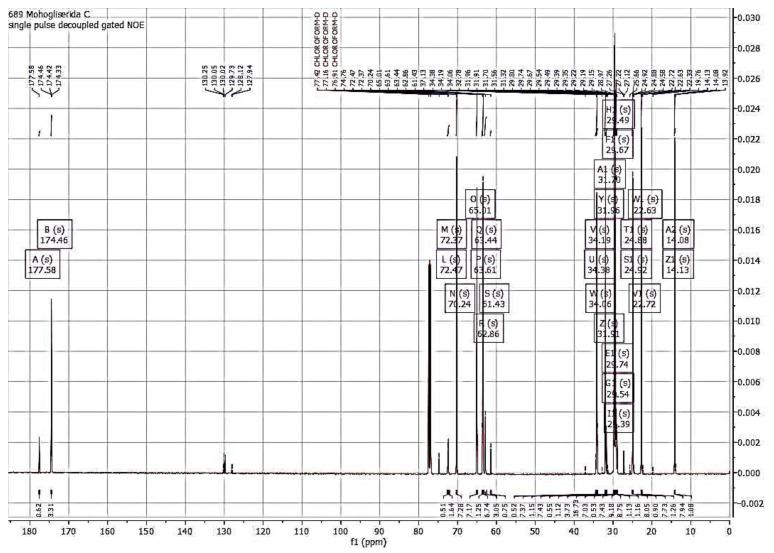
Spectrum ^13^C NMR of monolaurin from glycerolysis reaction of PKOo-PKS blend with a ratio of 40:60 *w*/*w* for 24 h, at 40 °C, a molar ratio of oil to glycerol 1:4, solvent to oil ratio of 2:1 *v*/*w*, enzyme concentration of 10% *w*/*w*, and stirring speed 600 rpm.

**Figure 5 foods-11-02412-f005:**
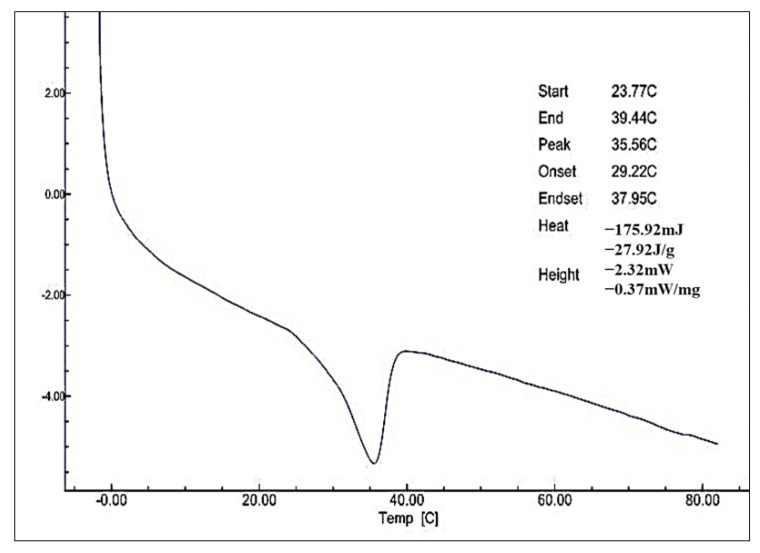
DSC Thermogram of monolaurin which was isolated from product glycerolysis palm kernel olein-stearin blend.

**Table 1 foods-11-02412-t001:** Physicochemical and emulsifying properties of monolaurin.

No	Parameter	Unit	Value
1	Saponification value	mg KOH/g	193.6 ± 0.83
2	Iodine value	wijs	0.82 ± 0.14
3	Melting point	°C	35.56
4	MAG	%	83.89 ± 5.52
5	Emulsion capacity (%)	%	93.66 ± 1.85
6	Emulsion stability	%	89.54 ± 3.36
7	HLB value		5.97 ± 0.06
8	Emulsion type		w/o

**Table 2 foods-11-02412-t002:** Inhibition zone of monolaurin from PKOo-PKS blend on *E. coli* FNCC 0091, *S. aureus* FNCC 0047, dan *B. subtilis* FNCC 0060 (mm).

Bacteria	Monolaurin Dose ** (ppm)	Solvent *	
		Aquadest	Alcohol
*E. coli* FNCC 0091	100	0.00 ± 0.00 c	6.00 ± 1.41c
	500	7.50 ± 0.00 b	8.83 ± 1.44 bc
	1000	8.00 ± 0.50 b	10.50 ± 2.29 bc
	2500	10.33 ± 0.76 ab	12.08 ± 3.09 ab
	5000	13.33 ± 2.36 a	13.67 ± 3.88 ab
*B. subtilis* FNCC 0060	100	7.15 ± 0.92 ab	9.00 ± 2.12 cd
	500	7.15 ± 0.92 ab	11.25 ± 1.06 c
	1000	7.50 ± 1.41 ab	12.25 ± 0.35 b
	2500	9.17 ± 1.26 a	12.50 ± 0.50 b
	5000	10.75 ± 1.56 a	16.75 ± 3.70 a
*S. aureus* FNCC 0047	100	0.00 ± 0.00 d	9.00 ± 1.41 c
	500	8.33 ± 2.36 c	9.67 ± 1.44 c
	1000	10.33 ± 2.75 c	10.17 ± 2.47 c
	2500	14.00 ± 1.50 b	13.33 ± 3.33 b
	5000	17.27 ± 1.54 a	15.17 ± 2.75 ab

* value is average ± SD (*n* = 3). A column’s average, followed by a different letter, indicates a significant difference in the Duncan Multiple Range Test (DMRT) 5%. ** The control treatment was aquadest alone with an inhibition zone 0.00 ± 0.00 mm.

## Data Availability

Data is contained within the article.

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
