# Peer review of "Enzymatic Glycerolysis of Palm Kernel Olein-Stearin Blend for Monolaurin Synthesis as an Emulsifier and Antibacterial"

_foods, 2022, doi:10.3390/foods11162412_

Round 1

Reviewer 1 Report

The authors in statistical analysis show samples were conducted with duplicates and in the table 2 value is average of n=3. What is the correct?

The conclusion should improve.

Erase line 396-397 (Glycerolysis………..RM) , line 402- 403 (with the formation…. Respectively). Line 404- 408 (Thermal analysis…… respectively).

The antibacterial activity with aquadest solvents has best inhibitory result and erase line 408-410 (has a minimum…. FNCC 0060).

Author Response

Response to Reviewer 1

[Comment-1] The authors in statistical analysis show samples were conducted with duplicates and in the table 2 value is average of n=3. What is the correct?

Response: Yes, it is correct. We have added an explanation of the samples in the statistical analysis section [page 4, statistical analysis section], as follow:

“………….with the exception on the antibacterial assay was conducted with triplicates”

[Comment-2] The conclusion should improve

Response: Thank you for your suggestion. We have improved the conclusion and deleted the first sentence in the conclusion and change with the sentence as follow: [page 12, conclusion section]

“Monolaurin can be synthesized from palm kernel olein-stearin blend and has the properties to be used as a food preservative and emulsifier. Monolaurin is also a valuable product due to its many biologically valuable properties and the benefits it provides to human health”.

[Comment-3] Erase line 396-397 (Glycerolysis…..RM), line 402-403 (with the formation…..Respectively). Line 404-408 (Thermal analysis…..respectively)

Response: we have removed the line suggested by the reviewer

[Comment-4] The antibacterial activity with aquadest solvents has best inhibitory result and erase line 408-410 (has a minimum…FNCC0060)

Response: We have erased line 408-410 (has a minimum…FNCC0060) and we have added  sentences [page 10, paragraph 2], as follow:

“The antibacterial activity with aquadest solvents has best inhibitory result at concentration of 100 ppm for B. subtilisFNCC 0060 and 500 ppm for E. coli FNCC 0091 and S. aureus FNCC 0047”

Reviewer 2 Report

Dear Author, I reviewed the manuscript (foods-1837134) entitled Enzymatic Glycerolysis of Palm Kernel Olein-Stearin Blend for Monolaurin Synthesis as an Emulsifier and Antibacterial. This manuscript presents relevant information about the emulsifier and antibacterial properties of monolaurin. However, some sections of the presented data can be improved. For this reason, I consider that this manuscript needs minor changes.

Additional comments.

Highlight the advantages of studying glycerolysis reactions to produce monolaurin and its potential applications in the food industry.

Check paragraph extension in this manuscript.

Try to include a bibliographical reference in the production and isolation of the monolaurin section. 

Include an experimental design that contains statistical factors and variables of response in the statistical analyses applied to the findings of this research.

Try to compare the obtained findings with similar assays where monolaurin bioactive properties were evaluated.

Try to include a monolaurin mode of action against the tested pathogenic bacteria.

Include future trends to keep working with the obtained data. 

Try to conclude with a general statement of the most relevant part of this study.

Author Response

Response to Reviewer 2

[General comment] This manuscript presents relevant information about the emulsifier and antibacterial properties of monolaurin. However, some sections of the presented data can be improved. For this reason, I consider that this manuscript needs minor change.

Response: Thank you very much

Some minor revisions for the authors to consider:

[Comment-1] Highlight the advantages of studying glycerolysis reactions to produce monolaurin and its potential applications in the food industry.

Response: Thank you for your suggestion. We have added a sentences at the end of paragraph 1 as follow :

“Monolaurin is a nontraditional antimicrobial agent with better antimicrobial activities but no health risks to consumers; however, its use as a preservative in the food industry is still limited” [Page 1, paragraph 1]

And at the end of paragraph 2 as follow:

“Monolaurin production from PKO through glycerolysis reaction. The advantage of glycerolysis process is simple to carry out, as fat or oil can be used directly as a substrate without the need to separate free fatty acids, so the glycerolysis reaction needs to be studied” [page 2, paragraph 2]

[Comment-2] Check paragraph extension in this manuscript

Response: The paragraph extensions in this manuscript have been checked

[Comment-3] Try to include a bibliographical reference in the production and isolation of the monolaurin section

Response: A bibliographical references in the production and isolation of monolaurin sections have been added [Page 2, paragraph 3], as follow:

“The separation of monoacylglycerol or monolaurin can be done using column chromatography with silica gel as the stationary phase and a mixture of solvents n-hexane and ethyl acetate which is gradually eluted [11,12], or solvent method using n-hexane and hydroalcoholic solution [13].”

[Comment-4] Include an experimental design that contains statistical factors and variables of response in the statistical analysis applied to the findings of this research

Response: We have added an experimental design in statistical analysis section [page 4, paragraph 3], as follows:

“A single factor completely randomized design (CRD) was used as experimental design”

[Comment-5] Try to compare the obtained findings with similar assays where monolaurin bioactive properties was evaluation

Response: We have added some sentences to explain the obtained finding with similar assays [page 11, at the end paragraph], as follow:

“Amin Zare et al. also reported that monolaurin dissolved in 95% ethanol had a minimal inhibitory concentration for E coli >4000 μg/ml while for S. aureus it was 128 μg/ml [39]. Sadiq et al. reported that the minimal inhibitory concentration for S. aureus ATCC 25923 was 100 μg/ml [40]. Wang et al. reported that the MIC for S. aureus and B. subtilis grown in nutrient broth was 1.25 mg/mL and 0.63 mg/mL, respectively, while for E. coli ≥ 10 mg/mL, respectively [41]. Buňková et al. report that monolaurin has an inhibitory effect on B. subtilis CCM 4062 and S. aureusCCM 3953 which are grown in nutrient broths with the addition of monolaurin at the lowest concentrations of 25 ppm and and 250 ppm, respectively, while in E. coli the inhibitory effect of monolaurin occurs at 1500 ppm [42]”

[Comment-6] Try to include a monolaurin mode of action against the tested pathogenic bacteria

Response: We have added some sentences to explain the obtained finding with similar assays [page 12, at the beginning paragraph], as follow:

“The mechanism of inhibition of monolaurin to E. coli is as follows: it first crosses the cell membrane and disrupts the normal functioning of DNA, eventually causing cell lysis. The effect of monolaurin activity on DNA is that it inhibits the DNA transcription process, causes a reduction in the synthesis of RNA and proteins, thus causing the cell cycle to stop and eventually causing inhibition of cell division [43]. The process of antimicrobial mechanisms includes the following three aspects: increased membrane permeability and cell lysis, disruption of the electron transport chain and oxidative phosphorylation separating, and inhibition of membrane enzymatic activity and nutrient uptake [44]”

[Comment-7] Include future trends to keep working with the obtained data

Response: We have added some sentences to explain the future trends  [page 12, paragraph 2], as follow:

“Based on this study, monolaurin have been used as an integral part of foodstuffs or as part of packaging material to improve food shelf life and/or inhibit microorganism growth. Trends in the future, consumers prefer naturally occurring antimicrobial com-pounds or preservatives to synthetic chemicals [41]”

[Comment-8] Try to conclude with a general statement of the most relevant part of this study

Response: We have deleted the first sentence in the conclusion and change with the sentence as follow: [page 12, conclusion section]

“Monolaurin can be synthesized from palm kernel olein-stearin blend and has the properties to be used as a food preservative and emulsifier”.

Reviewer 3 Report

Many thanks to the authors for their efforts in preparing this manuscript, but I have some comments;

In the antibacterial assay, you did not perform any control treatments ?

Figure (3); The peaks on the X-axis corresponding to the values between 3 and 5 overlaped with the numbers, replace the figure with a more clear one

In table 2, write the name of bacterial species in italic

Line 21: “MAG”, write the meaning of this abbreviation

Line 138: correct “menit” to “minutes”

Line 278, 291: correct ”PkO” to “PKOo”

Line 314: correct “Figure 3” to be “Figure 4”

Line 323: delete “analysis DSC”

Line 360: delete “both”

References:

Line 387, 388: The reference Nitbani et al did not take a number; it has been mentioned in the list of references No 2, 10, 27 and 35

Lines 434, 482, 486, 489, 514 an 515: write the names of organisms in italic

Reference NO 14 wrote in the text as “Solasea” and in thre reference list “Solaesa”

References NO 10 and 35 are duplicated

Author Response

Response to Reviewer 3

[General comment] Many thanks to the authors for their efforts in preparing this manuscript, but I have some comments:.

Response: Thank you very much

Some comment for the authors:

[Comment-1] in the antibacterial assay, you did not perform any control treatments?

Response: No, I have done control treatment with aquadest alone for antibacterial test with the results we have added below Table 2 [page 11] as follow:

“**The control treatment was aquadest alone with the inhibition zone 0.00 ± 0.00 mm”

[Comment-2] the peaks on the X-axis corresponding to the value between 3 and 5 overlaped with the numbers, replace the figure with a more clear one

Response: The Figure 3 has been changed [Page 8-9]

[Comment-3] In Table 2, write the name of bacterial species in italic

Response: The writing of bacterial species in Table 2 has been revised [page 11]

[Comment-4] Line 21: “MAG” write the meaning of this abbrevation

Response: We have explained the abbreviation of MAG [page 1], as follow:

…”monoacylglycerol (MAG)”…

[Comment-5] Line 138: correct “menit” to minutes

Response: We have revised “menit” to minutes [page 4]

[Comment-6] Line 278, 291: correct “PkO” to PKOo

Response: We have revised “PKO” to PKOo [page 8]

[Comment-7] Line 314: correct “Figure 3” to “Figure 4”

Response: We have revised “Figure 3” to “Figure 4” [page 9]

[Comment-8] Line 323: delete “analysis DSC”

Response: We have deleted “analysis DSC” in Figure 5 caption [page 9]

[Comment-8] Line 360: delete “both”

Response: We have deleted it [page 10]

[Comment-8] References: line 387, 388: the reference Nitbani et al did not take a number, it has been mentioned in the list of references No 2, 10, 27 and 35

Response: We have revised it [page 11]

[Comment-9] References: line 434, 482, 489, 514 and 515: write the names of organism in italic

Response: We have revised it [page 13-15]

[Comment-10] References No 14 wrote in text as ”Solasea” and in the reference list “Solaesa”

Response: We have revised it, the correct is “Solaesa” [page 6]

[Comment-11] References No 10 and 35 are duplicated

Response: We have revised it [page 13-14]

Reviewer 4 Report

 Enzymatic Glycerolysis of Palm Kernel Olein-Stearin Blend for Monolaurin Synthesis as an Emulsifier and Antibacterial

Consider writing in the title that monolaurin is a "high-value product". It not only has antibacterial and emulsifying properties but is also very valuable as a food supplement. It has a wide range of beneficial biological properties.

In the introduction, it is necessary for greater clarity to write the chemical equations of the reactions of hydrolysis and transesterification of coconut oil.

Writing the formulas of Lauric acid and methyl laurate will contribute to better visibility and make clear the exact conversion in the course of hydrolysis.

Ultimately, you have optimized the process, ie. you have selected optimal conditions for obtaining monolaurin by enzymatic hydrolysis - the most accurate temperature, enzyme concentration, and the most accurate equimolar coconut oil: glycerol ratios. This is a serious result, which is good to fix in the goal of your work because your development ends with serious success.

In conclusion, you again say that monolaurin can have the potential to be developed as a food preservative. Not only as a food preservative, monolaurin is also a product of high value due to its many biologically valuable qualities and the benefits it exerts on human health.

Your work is very interesting and valuable, I would recommend that you pay attention to the English language, and let your article be checked by an English-speaking editor.

Author Response

Response to Reviewer 4

[General comment] Considering writing in the title that monolaurin is a “high-value product”. It not only has antibacterial and emulsifying properties but is also very valuable as a food supplement. Its has a wide range of beneficial biological properties.

Response: Thanks for your comment. Our study in this paper just focused on the synthesis of monolaurin and its potential as an emulsifier and antibacterial. Research on monolaurin as a food supplement and its role for health is very interesting to develop and will be the focus of our research in the future. However we have added the sentence which explain that monolaurin is a high-valuable product [page 1, paragraph 1], as follow:

“Monolaurin is also a valuable product due to its many biologically valuable properties and the benefits it provides to human health”

[Comment-1] In the introduction it is necessary for greater clarity to write the chemical equations of the reactions of hydrolysis and transesterification of coconut oil

Response: thank you for your suggestion, but we thought it is not necessary to write the equations of hydrolysis and transesterification of coconut oil because raw material in this study is palm kernel oil.

[Comment-2] Writing the formula of lauric acid and methyl laurate will contribute to better visibility and make clear the exact conversion in the course of hydrolysis

Response:  Thank you for your suggestion. We have added the formula of lauric acid and methyl lauric in the sentence of paragraph 2 [page 1, paragraph 2], as follow:

“…..lauric acid (C12H24O2), methyl lauric (C13H26O2),….”

[Comment-3] Ultimately, you have optimized the process, ie. You have selected optimal conditions for obtaining monolaurin by enzymatic hydrolysis-the most accurate temperature, enzyme concentration, and the most accurate equimolar coconut oil:glycerol ratios. This is serious result, which is good to fix in the goal of your work because your development ends with serious success.

Response:  Thank you for your appreciation in our work.

Comment-4] In conclusion, you again say that monolaurin can have the potential to be developed as a food preservative. Not only as a food preservative, monolaurin is also a product of high value due to its many biologically valuable qualities and the benefits it exert on human health.

Response:  Thank you for your valuable and insightful comments. We have improved the conclusion and deleted the first sentence in the conclusion and change with the sentence as follow: [page 12, conclusion section]:

“Monolaurin can be synthesized from palm kernel olein-stearin blend and has the properties to be used as a food preservative and emulsifier. Monolaurin is also a valuable product due to its many biologically valuable properties and the benefits it provides to human health”

Comment-5] Your work is very interesting and valuable, I would recommend that you pay attention to the English language, and let your article be checked by an English-speaking-editor.

Response:  Thank you for your valuable and insightful comments that helped us improve this

manuscript. The manuscript was revised by an expertise in English language, following the sugestions.

Round 2

Reviewer 1 Report

The authors were the corrections.